# Impact of PSMA-PET/CT on Radiotherapy Decisions: Is There a Clinical Benefit?

**DOI:** 10.3390/cancers17081350

**Published:** 2025-04-17

**Authors:** Elías Gomis-Sellés, Antonio Maldonado, Miren Gaztañaga, Victoria Vera, Odile Ajulia, Gemma Sancho, Shankar Siva, Fernando Lopez-Campos, Felipe Couñago

**Affiliations:** 1Department of Radiation Oncology, Hospital Clínic Barcelona, 08036 Barcelona, Spain; 2Nuclear Medicine and Molecular Imaging Department, University Hospital Quironsalud Madrid/La Luz Hospital, 28003 Madrid, Spain; antonio.maldonado.suarez@gmail.com; 3Department of Radiation Oncology, San Carlos Hospital, Genesis Care Vithas La Milagrosa, 28010 Madrid, Spain; miren.gb@gmail.com; 4Department of Radiation Oncology, Badajoz University Hospital, 06006 Badajoz, Spain; vickyverab27@gmail.com; 5Nuclear Medicine Department, University Hospital Ramon y Cajal, 28034 Madrid, Spain; odileajuria@gmail.com; 6Department of Radiation Oncology, Hospital de la Santa Creu i Sant Pau, 08025 Barcelona, Spain; gsancho@santpau.cat; 7Peter MacCallum Cancer Centre, Sir Peter MacCallum Department of Oncology, University of Melbourne, Parkville 3010, Australia; shankar.siva@petermac.org; 8Department of Radiation Oncology, Hospital Universitario Ramón y Cajal, Genesis Care Vithas La Milagrosa, 28010 Madrid, Spain; fernando_lopez_campos@hotmail.com; 9Genesiscare España, Hospital Universitario San Francisco de Asis, Hospital Universitario La Milagrosa, Universidad Europea de Madrid, 28670 Madrid, Spain; felipe.counago@genesiscare.es

**Keywords:** PSMA, prostate cancer, radiotherapy, biochemical recurrence, metastasis-directed therapy, salvage radiotherapy, androgen deprivation therapy, target volume delineation, PET-based treatment planning

## Abstract

PSMA-PET/CT is changing prostate cancer management by significantly enhancing the accuracy of staging and treatment planning, particularly in radiotherapy. Its superior sensitivity over conventional imaging has led to more precise target delineation, optimized salvage radiotherapy strategies, and the identification of oligometastatic disease amenable to metastasis-directed therapy. As a result, PSMA-PET/CT frequently alters clinical decision-making, enabling more personalized treatment approaches that may improve disease control and patient outcomes. While its impact on oncologic management is undeniable, further prospective trials are needed to confirm its long-term clinical benefits and establish its definitive role in standard radiotherapy protocols.

## 1. Introduction

Prostate cancer (PCa) is the second most frequently diagnosed solid tumor in men and the fifth leading cause of cancer-related death worldwide in this population [1]. Advances in screening techniques and the widespread implementation of early detection strategies have significantly increased the diagnosis of localized disease, leading to improved survival rates. However, the rising prevalence of PCa has also made it a critical public health challenge [2].

Prostate-specific membrane antigen positron emission tomography/computed tomography (PSMA-PET/CT) is an important tool in prostate cancer staging. Its use has grown significantly in recent years due to its higher sensitivity than conventional imaging tests for detecting metastatic disease [3,4]. Moreover, some research groups have analyzed the detection capacity below the Phoenix criteria, presenting its potential usefulness [5].

In radiotherapy, where precision and personalization are essential, PSMA-PET/CT has ushered in a new paradigm, with one of its most significant contributions being its ability to shape and refine clinical decision-making. Studies have consistently demonstrated that PSMA-PET imaging frequently alters the management plan in men with BCR (biochemical recurrence; PSAnadir +2 ng/mL after radiotherapy or PSAnadir +0.2 ng/mL after prostatectomy) following definitive therapy. These changes include refining the delineation of radiotherapy volumes, initiating or deferring systemic treatments such as androgen deprivation therapy (ADT), or guiding focal interventions in oligometastatic disease [6,7]. Endpoints, such as freedom from androgen deprivation therapy, overall survival, and biochemical recurrence-free survival due to changes in patient management, are under research [8].

The clinical role of PSMA-PET imaging in managing prostate cancer is still under discussion. At the 2024 Advanced Prostate Cancer Consensus Conference (APCCC), an expert panel examined 36 questions regarding PSMA-PET imaging and its use in radiotherapy. Only 25% of these questions achieved consensus, indicating ongoing uncertainty and the need for additional evidence to standardize its clinical use [9].

Recognizing this knowledge gap, the primary purpose of this narrative review was to evaluate the documented impact of PSMA-PET on decision-making in radiation oncology and its clinical outcomes.

## 2. PSMA-PET/CT Radiotracers in Clinical Practice

The incorporation of PSMA-PET/CT into clinical practice has revolutionized the clinical management of prostate cancer. When referring to PSMA, we encompass a group of various radiopharmaceuticals that share the ability to bind to the extracellular domain of this transmembrane protein but, at the same time, exhibit different characteristics that may be beneficial in specific clinical scenarios.

The first PSMA radiopharmaceutical approved for clinical use, and therefore the one with the most evidence [10], is [^68^Ga]Ga-PSMA-11. It was developed by the Heidelberg group [11,12,13] and approved by the FDA in 2020 and the EMA in 2022. In a prospective study, Fendler et al. demonstrated a detection rate of 75% in patients with biochemical recurrence, with a mean PSA value of 2.1 ng/mL and a positive predictive value (PPV) of 92% [14]. The high detection rate of this radiopharmaceutical led to the development of other compounds, such as [^68^Ga]Ga-PSMA-617 and [^68^Ga]Ga-PSMA-I&T, which show similar biodistribution and comparable detection rates [15].

In recent years, several ligands bound to ^18^F have been developed, the most well-known being [^18^F]F-Piflufolastat, which was approved by the FDA and EMA in 2021 and 2023, respectively. Supported by studies such as OSPREY [16], this radiopharmaceutical demonstrated high sensitivity (95.8%) and PPV (81.9%) for disease detection, as well as the CONDOR study, which reported a high inter-reader agreement (Fleiss’ kappa: 0.65) in interpreting the scans [17].

Following the path of [^18^F]F-Piflufolastat, new ^18^F-bound compounds have emerged, such as [^18^F]F-PSMA-1007 and [^18^F]rhPSMA-7.3 [18]. Their detection rates are comparable to [^18^F]F-Piflufolastat, according to a systematic review by Huang [19]. Furthermore, in a prospective comparative study, [^18^F]F-Piflufolastat demonstrated a similar uptake in tumor lesions as [^18^F]F-PSMA-1007 [10].

PSMA radioligands mainly differ in the isotope they bind to and their biodistribution patterns. The most-used isotopes in hospital settings are [^68^Ga] and [^18^F] [20]. The half-life of [^18^F] is 110 min, which is longer than the ^68^ min of [^68^Ga], allowing for later image acquisition and thus achieving a better tumor-to-background uptake ratio (due to greater radiopharmaceutical washout and fewer non-specific bindings) [21,22]. Additionally, [^18^F] has lower energy (0.6 MeV) than [^68^Ga] (2.3 MeV), which translates into higher PET image resolution [21]. The synthesis of [^18^F] requires a cyclotron, whereas [^68^Ga] can be extracted from a ^68^Ge/^68^Ga generator, making it more accessible if such a generator is available at the center [20,23].

Regarding biodistribution, PSMA radiopharmaceuticals exhibit different organ uptake patterns, with the main difference being their route of elimination [24]. Unlike most compounds, [^18^F]F-PSMA-1007 is primarily eliminated through the hepatobiliary route, which favors the detection of local disease, particularly in patients with a history of prostatectomy, as one of the most common recurrence sites is the ureterovesical anastomosis [23,25,26]. On the other hand, the other mentioned compounds are primarily excreted via the urinary route, with a particular mention of [^18^F]rhPSMA-7.3, which has lower urinary excretion [18,19]. The use of diuretics or early dynamic or static imaging may be beneficial in such cases to mitigate the interference caused by radiopharmaceutical accumulation in the bladder when evaluating adjacent structures [20,24].

In advanced disease scenarios, the biodistribution of [^18^F]F-PSMA-1007 may be a drawback for assessing hepatic metastases [19], which should be considered when applying visual scales, such as the PSMAscore (requiring the spleen to be used as the reference organ instead of the liver for PSMAscore 2) [27]. It is important to note that in pilot studies, ^18^F-PSMA-1007 demonstrated superiority over ^18^F-FDG due to its high detection rate of prostate cancer lesions and excellent tumor uptake. Although non-tumor uptake in ^18^F-PSMA-1007 may lead to a potential misdiagnosis, recognizing these limitations and conducting a careful analysis can enhance diagnostic accuracy [28,29].

Another relevant radiotracer is ^68^Cu-PSMA. An observational study analyzed the bio-distribution of ^68^Cu-PSMA and compared it with ^68^Ga-PSMA. Both tracers had comparable distribution in intestines and exocrine glands; however, they differed in their excretion: ^68^Ga-PSMA is primarily excreted renally, while 64Cu-PSMA was excreted by hepatic and biliary pathways. Moreover, 64Cu had a significantly longer half-life (12 h vs. 68 min for ^68^Ga). This enables delayed imaging with improved contrast and makes it an alternative for centers without access to a near ^68^Ga generator [30]. Understanding radiopharmaceutical biodistribution is essential to avoid misinterpretations, such as uptake in the sympathetic chain, acute inflammatory processes, or indeterminate bone uptakes (UBU, for its English acronym) [26,31]. It has been determined that [^18^F]F-PSMA-1007 has the highest rate of UBU, highlighting the importance of recognizing these uptake patterns to prevent overstaging patients [19,32,33].

Despite the differences among PSMA compounds, no evidence supports the diagnostic superiority of any specific radioligand over others in terms of clinical impact [24].

Finally, we have to point out that patients with other histologies could also benefit from PSMA since it is mainly observed in tumor neovasculature [34]. This radiotracer has been studied, among other pathologies, in clear cell renal tumors, being considered by some authors as a valid option for diagnosis, staging, and response assessment [35]. It has also been studied with a teragnostic approach in glioblastoma in animal models [36].

## 3. Image Interpretation and Response Evaluation

### 3.1. The Role of PSMA-PET/CT in Radiotherapy Planning

The success of definitive therapy hinges on precise disease staging, particularly in identifying extra-prostatic extension and metastatic pelvic lymph node involvement. PSMA imaging plays a crucial role in accurately assessing disease extent and ensuring appropriate patient selection before radiation therapy for those undergoing curative-intent treatment, in this regard, PSMA-PET/CT has a relevant role (Figure 1, Figure 2 and Figure 3).

The OSPREY trial [16] highlighted the value of ^18^F-DCFPyL PSMA-PET/CT in the initial staging of high-risk prostate cancer, particularly for detecting nodal and distant metastases. The study reported a median positive predictive value of 86.7% (69.7–95.3%) and a negative predictive value of 83.2% (78.2–88.1%). Notably, PSMA-PET/CT often identifies metastatic spread beyond the conventional boundaries of pelvic lymph node dissection or radiotherapy fields, a finding observed in 16% to 48% of cases [37,38].

The multicenter randomized proPSMA trial [39] demonstrated that PSMA-PET/CT offers significantly greater accuracy than conventional imaging with a combined CT and bone scan (92% vs. 65%) for staging high-risk prostate cancer before curative-intent treatment, whether surgery or radiation. These findings support PSMA-PET/CT as a viable alternative to standard imaging for initial staging.

On the other hand, multiparametric Magnetic Resonance Imaging (mpMRI) is the preferred modality for defining intraprostatic targets and aiding radiotherapy planning. Studies comparing mpMRI and PSMA-PET/CT have shown similar sensitivity in delineating intraprostatic tumors, supporting their complementary roles in treatment planning. In this way, the combination of both techniques could be a good approach to improve tumor delineation. It demonstrated better accuracy than mpMRI for delineating intraprostatic gross tumor volume (GTV), with sensitivities and specificities of 86%/87% for PSMA-PET, 58%/94% for mpMRI, and 91%/84% for the combination [40,41,42,43].

The PRIMARY trial demonstrated enhanced accuracy when both techniques were used together, outperforming the targeted biopsy [44]. However, these studies are limited by small sample sizes and heterogeneous methodologies, underscoring the need for further research to refine GTV delineation.

Beyond staging, PSMA-PET/CT increasingly influences radiation therapy planning, altering treatment in 7–33% of cases [45,46]. Different cohort studies [7] reported treatment modifications in 13% of cases due to changes in TNM staging. A prospective evaluation of 73 patients with M0 localized prostate cancer (PCa) found that ^68^Ga-PSMA-11 PET-CT significantly influenced the intended definitive radiation therapy (RT) planning for PCa. The impact was noted in 16.5% (12/73) of patients whose elective RT fields included the prostate, seminal vesicles, and pelvic lymph nodes (LNs), and in 37% (25/66) of patients whose RT fields covered only the prostate and seminal vesicles, excluding the pelvic LNs [37].

While the long-term impact of these modifications remains unclear, the growing use of PSMA-PET/CT suggests it may soon become a routine part of PCa management, potentially influencing treatment decisions even before a biopsy.

### 3.2. PSMA-PET/CT in Salvage Radiotherapy

Conventional imaging methods, such as CT and bone scans, often fail to detect recurrence in cases of biochemical relapse after surgery (PSA > 0.2 ng/mL). As a result, salvage radiotherapy planning is typically based on probabilistic target volume delineation, focusing on the prostatic fossa as the most likely site of recurrence in the absence of visible disease. The integration of PSMA-PET/CT has significantly enhanced the detection of prostate cancer recurrence, offering superior sensitivity compared to standard imaging. Its ability to identify disease at lower PSA levels has the potential to refine target delineation in salvage radiotherapy, leading to more precise and personalized treatment planning.

The CONDOR trial (including 208 patients with BCR who underwent ^18^F-DCFPyL) [17] reported a disease detection rate of 73.3% at low PSA levels (<0.5) in cases where conventional imaging was negative. Similarly, a systematic review and meta-analysis (including 4790 patients) [47] found detection rates of 33% for PSA < 0.2 and 45% for PSA 0.2 to 0.5 ng/mL for ^68^Ga-PSMA.

The impact of PSMA-PET/CT on salvage radiotherapy (sRT) planning has been investigated in multiple studies, demonstrating its significant influence on treatment decisions. An observational study [48] analyzed 100 patients with biochemical recurrence (BCR) before radiotherapy, reporting a ^68^Ga-PSMA-PET/CT positivity rate of 76%, which led to treatment modifications in 59% of cases; 53.5% in the Koerber series [49]; and 60.5% in the Sterzing [45] series after radical prostatectomy.

One of the largest prospective multicenter studies included 420 patients, 312 with BCR. It found that ^68^Ga-PSMA-PET/CT influenced management in 62% of cases. Among men who planned for sRT, PET findings led to expanded treatment fields (12%), radiation boosts (15%), or reduced radiation dose/volume (4%) [50]. Another prospective study [51] evaluated 270 patients with BCR after prostatectomy and PSA < 1 ng/mL, finding that ^68^Ga-PSMA-PET/CT was positive in 49% of cases, predominantly detecting disease in bones or perirectal lymph nodes. This resulted in major management changes in 19% of patients, often involving expanded target volumes to include both the prostate bed and pelvic lymph nodes.

These differences have been observed with low levels of PSA [52], which demonstrated that in patients with low PSA < 0.5 ng/mL, ^68^Ga-PSMA-PET/CT altered treatment in 30% of cases. Moreover, an observational study [53] reported PET findings led to modifications in radiotherapy target volumes in 28% of post-prostatectomy patients. Overall, PSMA-PET/CT has been reported to impact management in 51–76% of BCR cases, with specific radiotherapy modifications occurring in 19–60% of patients. However, study limitations include heterogeneous patient populations with varying PSA levels at the time of imaging and a lack of detailed analysis of anatomic relapse patterns.

While PSMA-PET/CT agents play a significant role in guiding clinical management, their effect on overall survival in PCa patients remains uncertain. A recent prospective study [54] with a follow-up of 38 months (median) reported a three-year freedom from progression (FFP) of 64.5% (120/186) in men who underwent ^68^Ga-PSMA-11PET/CT imaging before sRT planning.

Prospective studies assessing long-term patient outcomes are essential, particularly for those in the earliest stages of biochemical recurrence who would otherwise go undetected by conventional imaging.

### 3.3. PSMA-PET/CT Imaging for Treatment of Oligometastatic Disease

PSMA-PET/CT has revealed the presence of oligometastatic disease, particularly in bone. Previously, this condition was only occasionally identified through conventional methods. Now, patients with limited metastatic spread are being considered for high-dose, targeted radiotherapy. This is a growing area of interest for radiation oncologists, as PSMA-PET/CT shows promise in detecting early oligometastatic disease and guiding treatment. However, its impact on patient outcomes remains uncertain (Figure 4).

Nevertheless, due to its increased sensitivity for early metastases, PSMA-PET/CT imaging has been incorporated into trials of MDT in PCa. The ORIOLE phase 2 randomized clinical trial [55] demonstrated that among men with oligometastatic PCa (up to three metastases), those receiving stereotactic ablative radiotherapy had a significantly lower risk of disease progression compared to those under observation alone (six-month progression rate of 19% vs. 61%). Similarly, in a prospective single-center study of 57 patients with oligometastatic PCa detected on PSMA-PET/CT, Kneebone and colleagues reported a median biochemical disease-free survival of 11 months following lesion-targeted stereotactic body radiotherapy without systemic treatment [56].

Ongoing research continues to explore the potential of PSMA-guided therapies. The STORM phase 2 randomized multicenter trial [57] is investigating whether combining whole pelvic radiation therapy with MDT offers a survival advantage over MDT alone in patients with pelvic nodal involvement. Additionally, the phase 2 SATURN trial [58] assessed the impact of SBRT guided by PSMA-PET alongside a six-month course of androgen deprivation therapy and dual androgen receptor pathway inhibitors. Among the 26 participants, 50% (13/26) maintained a PSA level below 0.05 ng/mL, with grades 2 and 3 AAT-related toxicities observed in 21% of patients.

The efficacy of PSMA-PET/CT guiding MDT was evaluated in a cohort of 42 patients with early oligometastatic castrate-resistant PCa who received radiation to all metastatic sites to delay systemic therapy initiation. This approach proved viable, with a median biochemical progression-free survival of 12 months and a median second-line systemic treatment-free survival of 15 months after PSMA PET-based MDT [59].

Although systemic androgen deprivation therapy remains the standard therapy for metastatic castrate-sensitive prostate cancer, regardless of lesion count, emerging clinical data support the integration of PSMA-PET/CT with MDT in this patient group. However, further investigation is needed to determine whether early detection and individualized radiation therapy translate into improved survival and quality of life.

## 4. PSMA-PET/CT in Biochemical Recurrence (BCR)

The most common site of prostate cancer recurrence is the lymph nodes, particularly regional chains (iliac, pre-sacral, and obturator). Extrapelvic lymph node metastasis occurs in 3–18% of cases, though isolated involvement is rare. Osseous structures are the third most common site, with 9–15% of patients affected [60].

In patients with BCR, imaging detects both local recurrences and distant metastases. However, conventional imaging (bone scan, CT) has low diagnostic yield in asymptomatic patients, as detection depends on PSA levels. The probability of a positive bone scan is <5% when PSA < 7ng/mL [61].

In this setting, PSMA-PET/CT is the imaging modality with the highest sensitivity at low PSA levels (<0.5 ng/mL), which is very far away from other molecular images (Table 1).

It may help distinguish patients with recurrences confined to the prostatic fossa from those with distant metastases, impacting the design and use of post-RP sRT [62,63,64,65,66].

Also, PSA alone is predictive of scan positivity. PSA doubling time is considered one of the strongest predictors of metastasis and PSA kinetics are preferred as a surrogate marker for the presence of recurrent compared to a single absolute PSA value. Some studies found detection rates over 83% and 93% for doubling times < 6.5 months [67].

The NCCN guidelines consider PSMA-PET/CT as the preferred test for whole-body staging of bone, soft-tissue, and visceral metastases in patients with biochemical recurrence after surgery or radiation therapy [68] (Table 2).

### 4.1. Local Recurrence

The most common site of recurrence after surgery is the prostatectomy bed, specifically at the uretero-vesical junction; this is identified in 39–71% of pelvic MRI studies [60].

After RT, MRI has shown excellent results in detecting local recurrences and guiding prostate biopsy [68]. Multiparametric MRI has high spatial resolution and superior tissue contrast resolution, which allows it to be the test of choice for imaging local recurrence after prostatectomy [68].

However, the analysis of prostatic fossa disease is a challenge for PSMA-PET/CT due to urinary excretion. Although some interventions suggested decreasing physiology urinary PSMA activity (i.e., early acquisition of PSMA-PET images as early as 6 min after radiotracer injection or administration of intravenous diuretics), MRI yields higher sensitivity for local recurrence (three times higher when PSA > 0.4 ng/mL) compared with PSMA-PET/CT [60].

Freitag et al. reported that only one-half of local recurrences seen at MRI after prostatectomy were identifiable at PSMA-PET/CT. The proximity of the recurrent lesion to the bladder was significantly associated with false negative PSMA-PET/CT results [69].

In another study, MRI had a better diagnostic performance for local recurrence than PSMA PET, with a sensitivity of 95% (vs. 74%) and an accuracy of 92% (vs. 78%) (Table 2) [70].

On the other hand, PSMA-PET/CT has the potential to guide and improve sRT planning. Defined gross disease within a target volume can be prescribed a higher dose. The clinical target volume (CTV) can be expanded to encompass areas of disease not seen by current first-line imaging and not normally targeted by consensus CTV [69].

A single-center, retrospective study involving 226 patients with biochemical relapse after prostatectomy and recurrence in the prostate bed (according to ^68^Ga-PSMA PET) found that 47% of relapses were uncovered or partially covered by CTV, according to RTOG guidelines. The posterior (48%) and posterolateral (29%) regions were mainly uncovered [71].

Moreover, the prospective randomized phase 3 PSMA-SRT NCT03582774 clinical trial demonstrated that PSMA-PET/CT findings can lead to major changes in local salvage radiotherapy in up to 33% of patients. This clinical trial included 193 patients with biochemical relapse after prostatectomy who underwent either salvage radiotherapy (control group, 90 patients) or PET/CT-guided salvage radiotherapy (Choline and PSMA were allowed). The aim was to assess the impact of PSMA-PET/CT on biochemical relapse and survival after radiotherapy, and the second endpoint was the impact of PSMA-PET/CT on radiotherapy planning. There was a 23% difference (*p* = 0.002) in the frequency of major changes between the control and PSMA arm and a 17.6% difference (*p* = 0.005) in the frequency of treatment escalation in the PSMA arm. The results of the main endpoint are not still available [72].

Another retrospective study selected 204 patients eligible for salvage radiotherapy. All patients underwent PSMA-PET/CT, with 109 showing persistent PSA and 5 having metastatic disease (both groups were excluded). Among the remaining 90 patients with biochemical recurrence, the scan detected the disease in 47% of cases. Radiotherapy treatment volumes were defined based on the scan results, covering the prostate bed, macroscopic tumor, pelvic lymphatic pathways, and/or affected pelvic lymph nodes. After a median follow-up of 23 months, biochemical recurrence-free survival was 78%, with no significant differences between PSMA-PET/CT-positive and -negative patients. Patients received androgen deprivation therapy before starting radiotherapy. Two patients experienced grade 3 genitourinary toxicity [73].

In this meta-analysis, the proportion of salvage RT prescribed with increased RT dose and/or increased target volume (dose escalation using SIB or enlarging target volumes) after PSMA-PET/CT increased to 24% [51].

The introduction of PET/CT imaging is just one of the significant developments that have begun to shape the care of patients with BCR, especially in distant recurrence [74]. MRI images could be helpful to guide biopsies or locate local recurrence [60]. Nowadays, the main guidelines recommend a PSMA-PET/CT, if available, in BCR after surgery or prostate radiotherapy [61,68,74].

### 4.2. PSMA-PET/CT in Metachronous Oligometastatic Disease and MDT

Oligometastatic prostate cancer (PCa) presents a clinical scenario that has become more frequent with the widespread use of PSMA-PET [75]. The superior accuracy of PSMA-PET/CT compared to conventional imaging has established it as one of the most powerful tools for detecting oligometastatic disease [39]. A recent survey conducted by the German radiation oncology group revealed that 97% of respondents prefer PSMA or Choline PET/CT to define oligometastatic disease [76]. Although clinical trials of MDT with this imaging test are limited, the evidence from cohort studies is large [77].

One of the largest series is the one published by Pastorello et al., where they collected a total of 95 patients with prostate cancer and 150 extraspinal bone lesions diagnosed by PSMA-PET/CT, achieving a local control of 89% at 3 years and with 5.3% intraosseous relapses. A BED equal to or greater than 198Gy was correlated with increased local progression-free survival (*p* = 0.007), with no major grade 2 toxicities. 51.5% of patients received concurrent hormone therapy [78].

Another observational single-institution study evaluated PSMA-PET-directed SBRT without initial ADT in 103 patients with oligometachronous prostate cancer (PCa) treated between 2014 and 2019. Patients received a median dose of 24 Gy in two fractions to bone metastases and 30 Gy in 3 fractions to lymph nodes, with time to biochemical failure as the primary endpoint. At a median follow-up of five years, 15% remained free of biochemical failure, and 28% were biochemically disease-free at the last follow-up after salvage treatments. Additionally, 39% of patients had never received ADT, and 55% remained ADT-free for relapse, with a median time to ADT initiation of 5.5 years. Two grade 3 toxicity events were reported (rib fracture and lymphoedema), and no local failures [75].

The study by Metz et al. analyzed the benefit of PSMA-PET/CT versus PET/CT-Choline in the treatment of patients with oligorecurrent prostate cancer. Between 2017 and 2020, patients who underwent radical prostatectomy and later experienced biochemical recurrence (PSA ≤ 2 ng/mL), diagnosed as oligometastatic using FCH or PSMA-PET/CT, were identified. The treatment approach included stereotactic body radiotherapy (SBRT), elective nodal or prostate bed radiotherapy with or without a boost, and androgen deprivation therapy (ADT) as needed. The primary endpoint was biochemical relapse-free survival (PSA increase ≥ 0.2 ng/mL above nadir and confirmed by two successive samples). The secondary endpoint was ADT-free survival. A total of 123 patients (70 with PSMA-PET/CT) were included, with a median follow-up of 42.2 months. The median biochemical relapse-free survival was 24.7 months for the PSMA group versus 13.0 months for the choline group (*p* = 0.008). ADT-free survival was also significantly longer for the PSMA group (*p* = 0.001). PSMA provided a clinical advantage over choline for guiding MDT in hormone-sensitive prostate cancer [79].

This potential clinical benefit of PSMA versus choline has been confirmed in the PRECISE-MDT Study. This retrospective study identified oligorecurrent prostate cancer patients with five or fewer nodal, bone, or visceral metastases detected by choline or PSMA-PET/CT who underwent SBRT, with or without systemic therapy, at eight tertiary cancer centers. The outcomes evaluated were progression-free survival (PFS), time to systemic treatment change due to polymetastatic conversion (PFS2), and overall survival (OS). After matching the propensity score, patients treated with PSMA-PET/CT-guided MDT showed longer PFS (HR = 0.49; *p* < 0.0001), PFS2 (HR = 0.42, *p* < 0.0001), and OS (HR = 0.39, *p* < 0.05) compared to those treated with choline PET/CT-guided SBRT [80].

We also found clinical trials of MDT based on PSMA. One of them is the phase II clinical trial by Glicksman et al. [81] where patients with biochemically relapsed prostate cancer after radical treatment, with negative conventional tests and no hormonal salvage therapy are selected. A total of 37 patients had an uptake on PSMA-PET/CT and received MDT with a follow-up of 15.9 months (median). The overall response rate was 60% including 22% no evidence of biochemical disease.

The five-year analysis of the TRANSFORM trial [82], which is one the largest cohort of men with oligometastatic prostate cancer (PCa) treated with SBRT-based metastatic-directed therapy (MDT) with 76.4% patients staging with PSMA-PET/CT. The primary endpoint was the five-year treatment escalation-free survival (freedom from new cancer therapy other than further SBRT). A total of 199 men received SBRT; 76.4% were hormone-naïve at baseline. The rate of five-year TE-FS was 21.7% (95% confidence interval [CI]: 15.7–28.7%) overall and 25.4% (95% CI: 18.1–33.9%) in the hormone-naïve subgroup. The subgroups with International Society of Urological Pathology Grade Groups 4–5 disease (hazard ratio [HR] = 1.48, 95% CI: 1.05–2.01, *p* = 0.026), a higher baseline prostate-specific antigen (PSA) (HR = 1.06, 95% CI: 1.03–1.09, *p* < 0.001), and those who received prior ADT (HR = 2.13, 95% CI: 1.40–3.26, *p* < 0.001) were at greater risk of treatment escalation. At a median follow-up of 67.9 months, 18.9% remained free from treatment escalation, and two had undetectable PSA levels. Finally, one of the most relevant trials about the clinical benefit of MDT according to PSMA-avid disease is the ORIOLE trial. It is a phase 2, randomized clinical study conducted at three centers to evaluate SBRT versus observation in men with 1–3 asymptomatic oligometastatic prostate cancer lesions. Eligible patients had metastases ≤5 cm, confirmed prostate cancer histology, and prior definitive treatment of the primary tumor. A total of 54 men were randomized 2:1 to SABR or observation. At six months, progression was significantly lower in the SBRT group (19% vs. 61%, *p* = 0.005), with improved median progression-free survival (not reached vs. 5.8 months, HR 0.30, *p* = 0.002). Total consolidation of PSMA-avid disease reduced the risk of new lesions (16% vs. 63%, *p* = 0.006). No grade ≥ 3 toxic effects were observed. The study followed CONSORT guidelines, with blinded radiologic assessments but unblinded treatment allocation. The treating physicians, participants, and data analysts were not blinded to treatment assignment. However, the trial radiologist evaluating response based on CT size criteria and ^18^F-DCFPyL uptake remained blind to both the treatment arm and the specific treatment fields [55].

Finally, Siva et al. analyzed 401 patients with 1–5 extracranial oligometastases treated with SBRT (prostate was the most common histology, 24%), assessing overall survival, progression-free survival, and the impact of total versus subtotal metastatic ablation. With a median follow-up of three years, the five-year OS and PFS rates were 54% and 14%, respectively, while 31% remained free from systemic therapy. Total metastatic ablation was associated with improved OS (HR 0.8, *p* = 0.032) and PFS (HR 0.6, *p* = 0.003), whereas medical operability did not impact outcomes. These findings support the complete ablation of all oligometastatic sites whenever feasible [83].

Although the largest sample size evidence is found in retrospective studies, there are clinical trials and large cohort studies of SBRT guided exclusively by PSMA (Table 3).

## 5. PSMA-PET/CT in the Scenario of Castration-Resistant Disease

PSMA-PET/CT has been used in CRPC in three different domains, conditioning changes in the therapeutic decision or selecting patients who might be more responsive to new treatments [84].

### 5.1. Selection of Oligometastatic Patients Who May Benefit from Metastasis-Directed Therapy

In this scenario, results from the randomized phase II ARTO trial [85] show that oligometastatic patients (less than four lesions, no visceral metastases) treated with SBRT in combination with abiraterone + prednisone had greater six-month PSA reduction and progression-free survival than patients who received standard treatment with abiraterone + prednisone only (HR:0.35 (95% CI, 0.21 to 0.57; *p* < 0.001) in the experimental versus control arm). However, this trial included both patients diagnosed with conventional tests (9.6%), Choline PET/CT (65.0%), and PSMA-PET/CT (25.4%).

At least one retrospective, observational, multi-institutional study [86] and one Phase I/II [87] have suggested that SBRT for oligoprogressive patients is an acceptable approach to delay the need for the next systemic treatment (NST) and avoid its side effects. In the first study, with a median follow-up of 23 months, the median NST-free survival was 13.1 months (95%CI 10.8–36.4). Grade ≥ 3 toxicity occurred only in 2 of 50 patients. In the study of Zhang et al. [87], 40% of patients were free of biochemical progression at 12 months and 21% at 24 months.

A review of five observational studies of mCRPC (up to five oligorecurrent or oligoproresistant lesions) diagnosed with PSMA-PET/CT and treated with metastatic targeted therapy and androgen depletion therapy analyzed the rates of overall survival, progression-free survival and survival to new lines of systemic treatment. Twenty-four patients were enrolled with a median follow-up of 33.8 months. Progression-free survival was 16.4 months (median), and progression-free survival to a new line of systemic therapy was 29 months (median). Median survival was not reached. Toxicities greater than or equal to grade 2 were 4.2%, and no grade 3 or higher toxicities were recorded [88].

### 5.2. Monitoring Disease Response to Systemic Therapy

Based on data from a multicenter retrospective study, a Response Evaluation Criteria in Prostate-specific Membrane Antigen PET/CT (RECIP 1.0) was developed to monitor treatment response in mCRPC patients undergoing PSMA-targeted radioligand therapy (PSMA-RLT) [89]. RECIP 1.0 integrates the appearance of new lesions and changes in PSMA-positive total tumor volume. In this study, PSMA-PET/CT images and follow-up of 124 patients were analyzed, and the authors observed that the disease response, stability, and progression states defined by RECIP criteria were associated with median survival (21.7, 13.1, and 8.3 months, respectively) and therefore established their prognostic value [90].

The second version of the prostate cancer molecular imaging standardized evaluation (PROMISE V2) [91] includes a reporting scheme for response parameters in clinical trials. It enables the application of existing PSMA-PET/CT response metrics, such as the PSMA-PET/CT Progression (PPP), focuses on the response of single lesions in PSMA-PET and the RECIP that relies on the PSMA-PET–derived total tumor volume and is more appropriate for extensive disease (Figure 5).

### 5.3. Selection of Optimal Treatment Candidates for PSMA-Targeted Radioligand Therapy (PSMA-RLT)

PSMA-PET has been used to select patients for inclusion in trials exploring the efficacy of Lu-PSMA in patients with mCRPC who have progressed to docetaxel and ARTA. In the case of the TheraP study [92], PSMA-PET/CT had to show at least one lesion with SUVmax ≥ 20 and no FDG-PET-positive and PSMA-negative lesions. In this study, treatment with Lu-PSMA was associated with longer progression-free survival versus cabazitaxel (HR 0.62; 95% CI 0.45–0.85). Men with a SUVmean ≥ 10 showed significantly higher odds of responding to 177Lu-PSMA-617 (vs. cabazitaxel) compared to those with a SUVmean < 10 (OR: 12.2, 95% CI: 3.4–59 vs. OR: 2.2, 95% CI: 1.1–4.5; *p* = 0.03). The authors did not observe any differences in overall survival between the randomized groups. However, patients excluded due to low PSMA expression or 2-[^18^F]FDG-discordant disease had a significantly shorter median overall survival [93].

In the VISION trial [94], patients with mCRPC who had received at least one ARTA and 1–2 taxane regimens were randomized to receive Lu-PSMA plus standard of care versus standard of care. Men had to have at least one PSMA-positive and no PSMA-negative lesions. Ad hoc analyses showed that patients with SUVmean ≥ 10.2 had a median rPFS and OS of 14.1 and 21.4 months, compared to 5.8 and 14.5 months for those with SUVmean < 6.0, respectively. According to the trial protocol, it is important to note that PSMA-positive lesions were defined as ^68^Ga-PSMA-11 uptake exceeding that of liver parenchyma in at least one metastatic lesion, regardless of size or location. PSMA-negative lesions were identified as those with PSMA uptake equal to or lower than liver parenchyma in lymph nodes with a short axis of ≥2.5 cm, metastatic solid-organ lesions with a short axis of ≥1.0 cm, or metastatic bone lesions with a soft-tissue component of ≥1.0 cm in the short axis.

These studies consistently show that patients with high mean SUV uptake experience better treatment responses to LuPSMA, while those with low or no uptake on PSMA-PET/CT tend to have poorer outcomes. Therefore, baseline PSMA-PET/CT findings, prior to radioligand therapy, can serve as a predictor of treatment response to radioligand therapy.

## 6. Current Trials

More than 250 active trials registered on clinicaltrials.gov provide a quantitative basis for understanding the impact of PSMA in prostate cancer management. We focus exclusively on those trials that may impact decisions regarding the radiotherapeutic treatment of prostate cancer and we highlight the following ones (Table 4).

In patients with biochemical recurrence and negative conventional imaging tests, study NCT03160794 [95] evaluates the role of whole-body MR in combination with PET PSMA to identify subclinical metastases and treat them with SABR.

### 6.1. Post-Prostatectomy Radiotherapy

The NCT03762759 [96] study has randomized patients with an indication for post-prostatectomy radiotherapy to fluciclovine F18 or ^68^Ga-PSMA PET to assess the potential impact on disease-free survival of using these radiotracers. The study also evaluates the decision to offer radiotherapy, to treat pelvic adenopathies, to dose escalate, CTV and PTV volumes, and dosimetric variables and their correlation with rectal and bladder toxicity.

The PERYTON study (NCT04642027) [97] is randomizing patients’ candidates for salvage radiotherapy of the prostatectomy bed to conventional vs. hypofractionated fractionation, guided by PET PSMA.

The PSMA relapsing study (NCT04794777) [98] is recruiting patients with an indication for salvage radiotherapy and randomizing them, following PSMA PET, to standard prostate bed irradiation or individualized RT treatment based on PSMA-PET/CT findings.

In MIDAS-Prostate (NCT05328505) [99] for patients with post-prostatectomy biochemical recurrence, the RT dose to the pelvis and/or bed is escalated or de-escalated depending on the PET PSMA uptake.

### 6.2. Oligometastases

In the setting of pelvic lymph node metastases diagnosed by PSMA, the OLIGOPELVIS2 study (NCT03630666) [100] is studying the role of pelvic salvage radiotherapy in combination with iADT (intermittent ADT).

The iSTOP study (NCT04619069) [101] evaluates the role of SBRT in hormone-sensitive stage IV patients. SBRT is performed on lesions that are negative by conventional imaging techniques but positive by PET PSMA.

The METRO study (NCT04983095) [102] randomizes hormonosensitive oligometastatic patients (de novo or recurrent) diagnosed by PSMA to the standard of care (3 years of ADT-abiraterone-prednisone + RT prostate and pelvis if the patient is oligometastatic de novo) to receive or not SBRT on PET PSMA-positive lesions. Its primary objective is failure-free survival.

Similarly, the METANOVA study (NCT06150417) [103] is randomizing patients with oligometastatic disease to receive SOC (standard systemic treatment + definitive local treatment on the primary) vs. SOC + RT on metastases. The diagnosis of metastases can be made with conventional imaging tests or with PET PSMA. Its primary objective is failure-free survival.

In the oligo-recurrence setting, the SPARKLE study (NCT05352178) [104] randomizes patients diagnosed with PSMA to metastasis-directed therapy (MDT) vs. MDT + one month of ADT vs. MDT + six months of ADT and enzalutamide.

Similarly, the ADOPT study (NCT04302454) [105] randomizes patients with oligorecurrence from PET PSMA to MDT ± 6 months of ADT, assessing the impact on metastasis-free survival.

The LUNAR study (NCT05496959) [106] has included patients with oligorecurrent disease treated with SBRT ± 177-Lutetium-PSMA. Patients are being followed with PSMA at periodic intervals to assess progression-free survival.

The DECREASE study (NCT04319783) [107] evaluates the role of consolidation radiotherapy (preferably SABR) on PSMA-detected lesions in patients with castration-resistant prostate cancer treated with Darolutamide.

### 6.3. Future Directions

PSMA-PET/CT is reshaping the management of prostate cancer, especially in the context of radiotherapy. Its ability to detect lesions with greater sensitivity and specificity than conventional techniques has made it a key tool for treatment planning.

It is expected to allow us in the near future to:1.Further improve staging and patient selection;2.Personalize radiotherapy (doses, volumes);3.Standardize radiotherapy aimed at metastases with ablative techniques;4.Monitor response and guide re-irradiation techniques;5.Integrate with new therapeutic approaches such as radiopharmaceuticals.

PSMA-PET/CT is transforming the use of radiotherapy in prostate cancer, allowing for greater precision in patient selection, dose optimization, and better disease control. As the evidence and availability of this technology expand, its role in radiotherapeutic planning is likely to become standard in clinical practice, improving oncological outcomes and patients’ quality of life.

There are other critical operational challenges to consider. On the one hand, accessibility to these radiotracers is limited, and costs may be a barrier [108]. ^18^F-PSMA-1007 is a very practical radiotracer for prostate cancer imaging due to its high throughput, transportability, and specificity in detecting recurrences. In contrast, ^68^Ga-PSMA-11 must be produced in situ due to its short half-life and lower yield, although production costs are similar [109]. Although ^68^Ga-PSMA-11 PET in biochemical recurrence is related with less than 10% false positives (post-radiotherapy prostatic uptake being the most relevant, originating from treated benign tissue or potentially indolent tumor remnants) [110]. On the other hand, the interdisciplinary integration of PSMA-PET/CT is complex and requires protocols and standardized guidelines for image interpretation. With these objectives in mind, the PROMISE criteria have emerged to provide a standardized way of reporting the disease, thus facilitating clinical decision-making and objectifying the oncological response [91].

## 7. Conclusions

PSMA-PET/CT has emerged as a transformative imaging modality in the management of prostate cancer, significantly impacting radiotherapy decision-making. Its superior sensitivity and specificity enable more precise staging, treatment planning, and response assessment, leading to improved personalization of radiotherapy strategies. Integrating PSMA-PET/CT into clinical workflows has shown promise in optimizing patient selection, refining target volumes, and guiding metastasis-directed therapies. While PSMA-guided SBRT clinical benefits are increasingly evident, further prospective trials are necessary to quantify its long-term impact. As evidence continues to accumulate, PSMA-PET/CT is poised to become a cornerstone in the standard of care for prostate cancer radiotherapy.

## Figures and Tables

**Figure 1 cancers-17-01350-f001:**
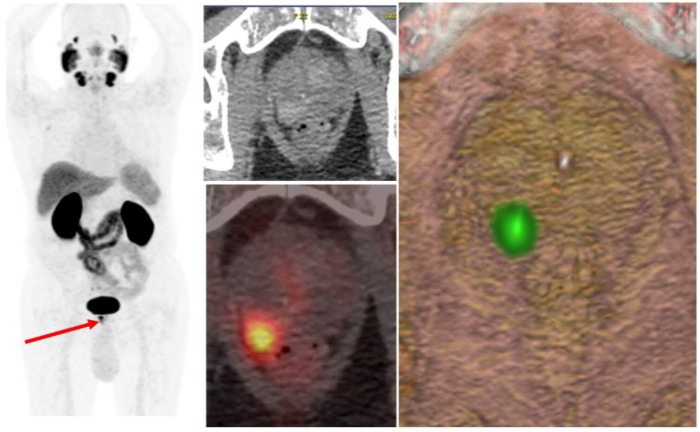
Clinical case of a local recurrence. PSMA-PET/CT patient with PSA of 13 ng/mL and an adenocarcinoma of the prostate, Gleason score 8 (4 + 4), classified as high-risk. PSMA-PET/CT revealed a tumor lesion located in the posterior peripheral zone of the right prostatic lobe (red arrow), with obliteration of the right rectoprostatic recess suggestive of extraglandular extension (cT3a). No evidence of nodal or distant metastases was identified (cN0 cM0).

**Figure 2 cancers-17-01350-f002:**
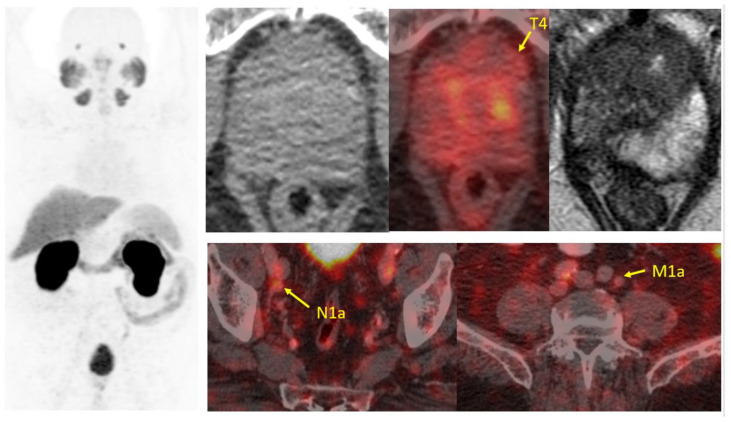
Clinical case of diagnosis with PSMA-PET/CT. A patient diagnosed with prostate adenocarcinoma, Gleason score 7 (4 + 3). PSMA-PET/CT staging revealed evidence of locally advanced disease, corresponding to a molecular stage of T4 N1a M1a (miTNM v1.0) (yellow arrows). There is an adenopathy in the right ilio-obturator chain (7.2 × 8.5 mm) with an SUV of 3.4 and a PSMA score of 1. Prostatic hyperplasia is observed with a heterogeneous uptake of the biomarker. The area of greatest activity is located in the left apical segment (SUV 9.4/PSMA score 1), extending toward the anterior margin of the left puborectalis muscle. Additionally, a lymph node is observed in the left common iliac chain, 9.4 × 9.4 mm, with an SUV of 4.1 and a PSMA score of 1.

**Figure 3 cancers-17-01350-f003:**
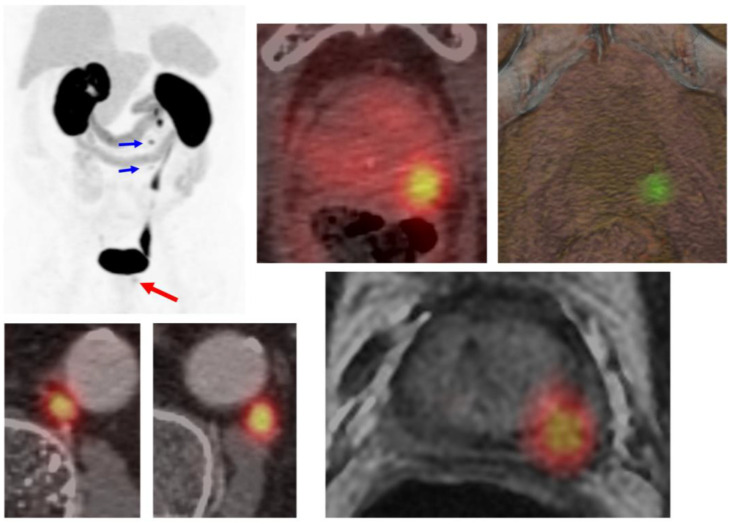
Local and regional recurrence prostate cancer. A patient with prostate cancer, clinical stage cT3a N1 M0, Gleason score 7 (3 + 4), and initial PSA of 4 ng/mL. Treated with radiotherapy and hormonal therapy (RT-HT), achieving a PSA nadir of 0.5 ng/mL. Subsequent PSMA-PET/CT identified disease recurrence in the peripheral zone of the prostate (red arrow) and retroperitoneal lymph nodes (blue arrows).

**Figure 4 cancers-17-01350-f004:**
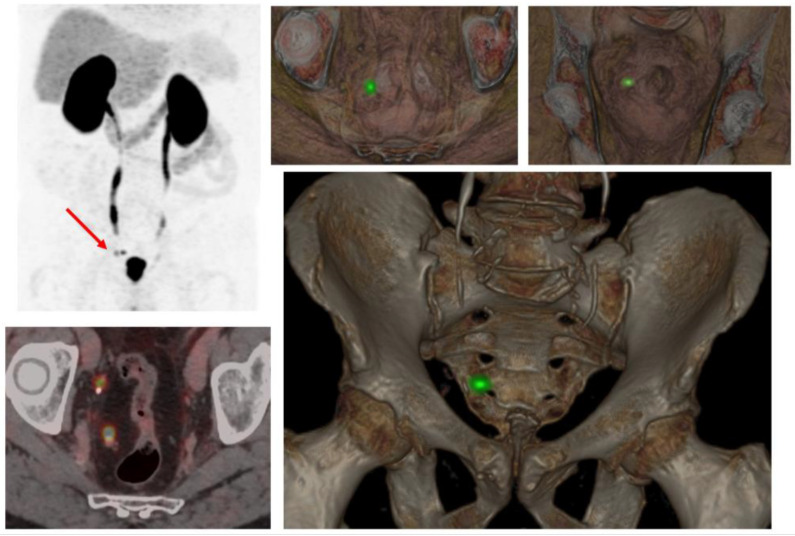
Clinical case of regional recurrence. A patient with prostate cancer, Gleason score 8, was initially treated with radical prostatectomy followed by adjuvant radiotherapy. Subsequent PSA elevation to 0.35 ng/mL prompted PSMA-PET/CT, which revealed a recurrence in a right pararectal lymph node (red arrow).

**Figure 5 cancers-17-01350-f005:**
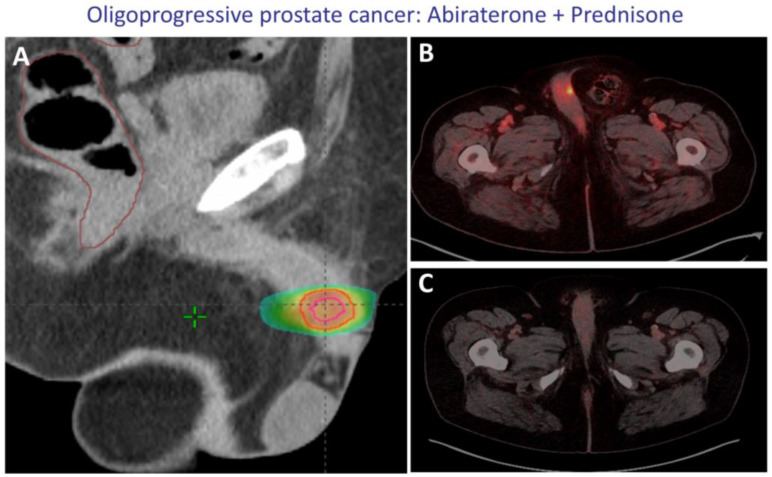
Real patient treated at HSCSP. (**A**) SGRT-SBRT-arctherapy; 30 Gy, every other day. The treatment volume is shown in pink and red outline together with the isodose curves. It can be seen that the dose is shaped by the treatment region. (**B**) CRPC patient on treatment with abiraterone + prednisone progressing in a single site at the level of the corpus cavernosum (focal PSMA uptake). (**C**) PSMA complete remission four months after SBRT and PSA < 0.03 ng/mL.

**Table 1 cancers-17-01350-t001:** Comparative detection rates (DR) of choline PET/CT and PSMA-PET/CT at different PSA levels [62,63,64,65,66].

Level PSA (ng/mL)	^68^Ga-PSMA-PET (DR)	Choline-PET/CT (DR)
<0.2	33%	5–24%
0.2–0.49	45%	5–24%
0.5–0.99	59%	36%
1.0–1.99	75%	43%
>2.0	95%	62%
>3.0	>95%	73%

**Table 2 cancers-17-01350-t002:** Differences between sensitivity (S), specificity (SE), and accuracy (A) in the diagnosis of biochemical recurrence with different imaging techniques [68].

	PSMA-PET/CT	PSMA-PET/CT	PSMA-PET/CT	mpMRI	mpMRI	Bone Scintigraphy
	Local Recurrence	Lymph Nodes	Bone Metastases	Local Recurrence	Lymph Nodes	Bone Metastases
Sensitivity	63%	83.3%	83.3%	90.9%	41.7%	50.0%
Specificity	73.7%	80%	92%	94.7%	94.4%	84%
Accuracy	77.8%	90.6%	71.0%	92.3%	72%	77.4%

**Table 3 cancers-17-01350-t003:** Key prospective trials evaluating PSMA-guided SBRT in prostate cancer.

Study	Design	N	Concurrent Hormonal Treatment	Median Follow-Up (Months)	Results
Mohan et al. [75]	Cohort single-institution.	103	Not allowed.	60.0	Biochemical failure (BF, nadir + 0.2) was 1.1 years (0.90–1.3). 2-year BF free survival was 25% (18–35%). 5-year BF free survival was 15% (9.2–25%)
Glicksman et al. [81]	Prospective single-institution. Phase II	72 (53% had PSMA-detected oligorecurrent disease)	Not allowed.	15.9	The overall response rate was 60%
TRANFORM trial [82]	Cohort single-institution.	199	Yes (7.0%) Prior ADT (16.6%)	67.9	At the end of follow-up, 18.9% of patients were free from treatment escalation. Treatment escalation-free survival after SBRT: 51.7% (2-year), 21.75 (5-year).

**Table 4 cancers-17-01350-t004:** Current trials.

Trial Name	Clinical Scenario	Intervention	Phase	N	Status	NCT Identifier
[95]	Biochemical failure	PSMA/MR + SABR	II	100	Recruiting	NCT03160794
RAD4516-18 [96]	Post-prostatectomy	Fluciclovine F18 or ^68^Ga-PSMA-PET/CT for planning	II	140	Active, not recruiting	NCT03762759
PERYTON [97]	Post-prostatectomy	Conventional vs. Hypofractionated RT	III	538	Recruiting	NCT04642027
PSMA recidiv [98]	Post-prostatectomy	Salvage RT vs. PSMA targeted RT	III	450	Recruiting	NCT04794777
MIDAS-Prostate [99]	Post-prostatectomy	RT dose modulation per PSMA	II	80	Recruiting	NCT05328505
OLIGOPELVIS2 [100]	Oligometastatic (pelvic nodes)	Intermittent ADT ± pelvic RT	III	256	Active, not recruiting	NCT03630666
iSTOP [101]	Oligometastatic	Intermittent ADT ± SBRT	I/II	30	Active, not recruiting	NCT04619069
METRO [102]	Oligometastatic	SOC ± MDT	III	118	Recruiting	NCT04983095
METANOVA [103]	Oligometastatic	SOC ± MDT	II	200	Recruiting	NCT06150417
SPARKLE [104]	Oligorecurrent	MDT ± 1mo ADT or 6 mo ADT/enza	III	873	Recruiting	NCT05352178
ADOPT [105]	Oligorecurrent	MDT ± 6 mo ADT	III	280	Recruiting	NCT04302454
LUNAR [106]	Oligorecurrent	SBRT ± 177-Lutetium-PSMA	II	93	Active, not recruiting	NCT05496959
DECREASE [107]	mCRPC	Darolutamide ± RT	II	70	Recruiting	NCT04319783

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
