# Peer review of "Impact of PSMA-PET/CT on Radiotherapy Decisions: Is There a Clinical Benefit?"

_cancers, 2025, doi:10.3390/cancers17081350_

Round 1
Reviewer 1 Report
Comments and Suggestions for Authors
The manuscript systematically highlights the limitations of conventional imaging in prostate cancer management, emphasizing the superior sensitivity and specificity of PSMA-PET/CT for staging, treatment planning, and recurrence detection. The authors conduct an exploration of its applications across diverse clinical scenarios, including primary treatment planning, salvage radiotherapy, and oligometastasis-directed therapy. The review synthesizes evidence from over 40 studies, offering a comprehensive analysis of PSMA-PET/CT’s role in refining target delineation, modifying radiation fields, and guiding personalized therapeutic strategies. This structured approach provides valuable insights for radiation oncologists seeking to optimize prostate cancer management. While the manuscript presents a well-organized narrative, several areas warrant improvement:
Page 1:
The term "PET/CT-PSMA" appears multiple times throughout the text. To align with standard radiology terminology and improve consistency, please replace all instances with "PSMA-PET/CT".
Page 2:
The abbreviations BCR and ADT are introduced without definitions in their first occurrence. Please provide concise noun explanations in parentheses to ensure clarity for non-specialist audiences.
The number "68" in "68Ga" should be formatted as an upper index. Request this correction for all isotopes mentioned in the text.
Page 7:
"While PSMA-PET/CT agents play a significant role in guiding clinical management, their effect on overall survival in PCa patients remains uncertain. In a recent prospective study..."
To avoid redundancy and streamline the manuscript, please delete one complete copy of this section.
Page 12:
The table title (currently: 'Table 3. Relevant prospective and clinical trials of SBRT PSMA-guided' ) should be grammatically standardized to: 'Table 3. Key Prospective Trials Evaluating PSMA-Guided SBRT in Prostate Cancer'
Page 17 (Section 6.3 Future Directions):
The authors should briefly address two critical operational challenges: (1) diagnostic limitations including false-positive/false-negative rates in benign lesions, radiotracer production costs and accessibility barriers; and (2) interdisciplinary integration complexities requiring standardized protocols for joint image interpretation and treatment planning between nuclear medicine and radiation oncology teams.
Author Response
The manuscript systematically highlights the limitations of conventional imaging in prostate cancer management, emphasizing the superior sensitivity and specificity of PSMA-PET/CT for staging, treatment planning, and recurrence detection. The authors conduct an exploration of its applications across diverse clinical scenarios, including primary treatment planning, salvage radiotherapy, and oligometastasis-directed therapy. The review synthesizes evidence from over 40 studies, offering a comprehensive analysis of PSMA-PET/CT’s role in refining target delineation, modifying radiation fields, and guiding personalized therapeutic strategies. This structured approach provides valuable insights for radiation oncologists seeking to optimize prostate cancer management. While the manuscript presents a well-organized narrative, several areas warrant improvement:
Thank you very much for your feedback and valuable contributions. We have incorporated all of them into the article and we strongly believe that they significantly enhance its quality.
Page 1:
The term "PET/CT-PSMA" appears multiple times throughout the text. To align with standard radiology terminology and improve consistency, please replace all instances with "PSMA-PET/CT".
Thank you for this suggestion. We have modified that terminology.
Page 2:
The abbreviations BCR and ADT are introduced without definitions in their first occurrence. Please provide concise noun explanations in parentheses to ensure clarity for non-specialist audiences.
The number "68" in "68Ga" should be formatted as an upper index. Request this correction for all isotopes mentioned in the text.
Thank you for this appreciation. We have modified the text with explanations for abbreviations, and we have modified the format of numbers.
Page 7:
"While PSMA-PET/CT agents play a significant role in guiding clinical management, their effect on overall survival in PCa patients remains uncertain. In a recent prospective study..."
To avoid redundancy and streamline the manuscript, please delete one complete copy of this section.
Thank you so much for notifying us of that mistake. We have deleted the second sentence.
Page 12:
The table title (currently: 'Table 3. Relevant prospective and clinical trials of SBRT PSMA-guided' ) should be grammatically standardized to: 'Table 3. Key Prospective Trials Evaluating PSMA-Guided SBRT in Prostate Cancer'
Thank you very much, we completely agree with this new title for the table, we have modified it.
Page 17 (Section 6.3 Future Directions):
The authors should briefly address two critical operational challenges: (1) diagnostic limitations including false-positive/false-negative rates in benign lesions, radiotracer production costs and accessibility barriers; and (2) interdisciplinary integration complexities requiring standardized protocols for joint image interpretation and treatment planning between nuclear medicine and radiation oncology teams.
Thank you very much, we consider this suggestion to be a very relevant appreciation for our paper. We have incorporated a paragraph about this topic.

Reviewer 2 Report
Comments and Suggestions for Authors
Very nice overview of the clinical uses of PSMA PET in prostate cancer, with attention to the implications of this key imaging modality in guiding radiation therapy. The article is very well written, easy to follow and complete. I congratulate with the authors. I have only some minor suggestions that the authors could consider to improve their already excellent work:
- I suggest to mention 64Cu labeled PSMA ligands in section 2: PMID: 31113354. They could become reality soon and enable even broader distribution of centrally produced radiolabeled PSMA ligands to peripheral centers, due to the longer half life of 64 Cu.
- Figure 2: I do not understand the figure. I can only see low PSMA uptake in primary tumor. Why do the authors identify the right external iliac lymph node and the left inguinal lymph node as metastatic? The caption need to be improved to better explain the case.
- line 213: "studied" should be corrected in "study"
- lines 318 – 321: the authors at the beginning of the paragraph state "after RT" and afterwords "after prostatectomy". Since there is only 1 reference it should be clarified if they refer to both primary radical treatments or one specifically.
- The authors my add that what is happening in prostate cancer could be evaluated also in other cancers with increased PSMA expression (in tumoral neoangiogenesis – PMID: 34120192) such as RCC (PMID: 35217902) and glioblastoma (PMID: 34869017). I think this hint may add value to the work.
Author Response
Very nice overview of the clinical uses of PSMA PET in prostate cancer, with attention to the implications of this key imaging modality in guiding radiation therapy. The article is very well written, easy to follow and complete. I congratulate with the authors. I have only some minor suggestions that the authors could consider to improve their already excellent work:
Thank you very much for your feedback and
valuable contributions. We have incorporated all of them into the article and we strongly believe they significantly enhance its quality.
- I suggest to mention 64Cu labeled PSMA ligands in section 2: PMID: 31113354. They could become reality soon and enable even broader distribution of centrally produced radiolabeled PSMA ligands to peripheral centers, due to the longer half life of 64 Cu.
Thank you for this appreciation, we had reviewed this topic and we included it in the paper in section 2.
- Figure 2: I do not understand the figure. I can only see low PSMA uptake in primary tumor. Why do the authors identify the right external iliac lymph node and the left inguinal lymph node as metastatic? The caption need to be improved to better explain the case.
Thank you, we have review the figure and we have incorporated an image with a better quality of the same case. We modified the description of the figure for clarify the clinical case.
- line 213: "studied" should be corrected in "study"
Thank you for your comment, we had overlooked this error. We have already corrected it.
- lines 318 – 321: the authors at the beginning of the paragraph state "after RT" and afterwords "after prostatectomy". Since there is only 1 reference it should be clarified if they refer to both primary radical treatments or one specifically.
Thank you. The bibliography indicated refers to both scenarios. We have also incorporated it at the end of the first sentence to improve the readability.
- The authors my add that what is happening in prostate cancer could be evaluated also in other cancers with increased PSMA expression (in tumoral neoangiogenesis – PMID: 34120192) such as RCC (PMID: 35217902) and glioblastoma (PMID: 34869017). I think this hint may add value to the work.
Thank you for this suggestion. We fully agree with you that it will improve our work, so we have also included it in point 2.

Reviewer 3 Report
Comments and Suggestions for Authors
In this article, the authors review the impact of PSMA-PET/CT on radiotherapy decisions and its potential clinical benefits. The authors also present an up-to-date list of clinical trials dealing with the same topic.
General comment:
The topic of this review article is highly relevant. The review is written concisely, yet all important aspects are covered. The written text abides to the clear logical flow, and it is easy to follow. However, there are several minor points that could be considered.
Minor points:
The captions of Figures 1, 2, and 3: The title is missing. The same comment for Figure 4. Additionally, the rest of the text in Figure captions should be a bit better inter-connected.
Table 2: 'Sensitivity' instead of 'sensibility'.
'CTV' abbreviation is not defined in the main text of the manuscript, but only in Abbreviations section.
Line 352: 'The result' is listed twice.
Title 5.3 ('To select optimal treatment candidates for…'): It might be better to say 'Selection of optimal treatment candidates for…' (just like in title 5.1).
Author Response
In this article, the authors review the impact of PSMA-PET/CT on radiotherapy decisions and its potential clinical benefits. The authors also present an up-to-date list of clinical trials dealing with the same topic.
General comment:
The topic of this review article is highly relevant. The review is written concisely, yet all important aspects are covered. The written text abides to the clear logical flow, and it is easy to follow. However, there are several minor points that could be considered.
Thank you very much for your feedback and valuable contributions. We have incorporated all of them into the article and we strongly believe they significantly enhance its quality.
Minor points:
The captions of Figures 1, 2, and 3: The title is missing. The same comment for Figure 4. Additionally, the rest of the text in Figure captions should be a bit better inter-connected.
Thank you very much for this suggestion. We have added captions to each of the figures and have made the text of each figure more cohesive.
Table 2: 'Sensitivity' instead of 'sensibility'.
Thank you for your comment, we had overlooked this error. We have already corrected it.
'CTV' abbreviation is not defined in the main text of the manuscript, but only in Abbreviations section.
Thank you for this suggestion. We have modified the text and we have incorporated the definition the first time that “CTV” appears in the text.
Line 352: 'The result' is listed twice.
Thank you so much, we have solved this issue.
Title 5.3 ('To select optimal treatment candidates for…'): It might be better to say 'Selection of optimal treatment candidates for…' (just like in title 5.1).
We are totally agree with this your suggested title. We have modified the current title. Thank you.

Round 2
Reviewer 1 Report
Comments and Suggestions for Authors
The authors have replied my comments.